# Gorham-Stout Disease with Multiple Bone Involvement—Challenging Diagnosis of a Rare Disease and Literature Review

**DOI:** 10.3390/medicina57070681

**Published:** 2021-07-02

**Authors:** Alina Momanu, Lavinia Caba, Nicoleta Carmen Gorduza, Oana Elena Arhire, Alina Delia Popa, Victor Ianole, Eusebiu Vlad Gorduza

**Affiliations:** 1Neuromotor Recovery Department, Rehabilitation Hospital Iasi, 700661 Iasi, Romania; malyna.2004@yahoo.com; 2Department of Medical Genetics, Faculty of Medicine, “Grigore T. Popa” University of Medicine and Pharmacy, University Street, No 16, 700115 Iasi, Romania; vgord@mail.com; 3Endocrinology Department, “St. Spiridon” Hospital, 700111 Iasi, Romania; cargorduza@yahoo.fr; 4Arhimed Medical Center, 700613 Iasi, Romania; oanaarhire@yahoo.com; 5Nursing Department, “Grigore T. Popa” University of Medicine and Pharmacy, University Street, No 16, 700115 Iasi, Romania; roxyal04@gmail.com; 6Pathology Department, “Grigore T. Popa” University of Medicine and Pharmacy, University Street, No 16, 700115 Iasi, Romania; ianole.victor@gmail.com

**Keywords:** Gorham-Stout disease, osteolysis, spine lesions

## Abstract

Gorham-Stout disease is a rare disorder, which may result in a poor prognosis. This disease, a rare lymphangiomatosis, is defined by progressive bone disappearance due to massive unicentric and multicentric osteolysis. Osteolytic lesions of the spine and pleura effusion are poor prognostic factors. Herein, we will present a case where the onset of disease occurred at the age of 18 with asthenia, myalgia, and major bone pain, followed by incomplete motor deficiency in the lower limbs and, later, in the upper limbs. Imaging studies (CT scan and MRI) of the patient revealed osteolytic lesions (cervical and thoracic vertebrae, rib, and clavicle) and a pathological fracture of the C7 vertebra. Surgical procedures undertaken involved replacing the affected vertebrae with bone grafting and prosthesis. The investigations performed allowed for the exclusion of inflammation, thyroid or parathyroid disease, lymphoma, neoplasia, or autoimmune disorders. A bone marrow biopsy showed osteolysis, the replacement of bone tissues with connective tissue, and chronic non-specific inflammation. The evolution was negative with almost complete osteolysis of the left clavicle, the emergence of new osteolysis areas in the lumbar vertebrae, pelvic bones, and the bilateral proximal femur, splenic nodules, chylothorax, and associated major neurological deficits. Unfortunately, this negative evolution resulted in the patient’s death a year after onset.

## 1. Introduction

Gorham-Stout disease (GSD), also known as Gorham–Stout syndrome, massive osteolysis, vanishing bone disease, phantom bone disease, and Gorham’s disease, is a very rare condition characterized by massive osteolysis and lymphangiomatosis. Approximately 300 such cases were described in the literature [1]. The first case was described in 1838 by Jackson, but it was Gorham and Stout, in 1955, that defined the main elements of the condition: osteolysis and lymphangiomatosis [2]. GSD represents one of the idiopathic osteolysis according to Hardegger’s classification: type 1—hereditary multicentric osteolysis with dominant transmission, type 2—hereditary multicentric osteolysis with recessive transmission, type 3—non-hereditary multicentric osteolysis with nephropathy, type 4—Gorham’s massive osteolysis, and type 5—the Winchester Syndrome [3]. Recently, in the classification of The International Society for the Study of Vascular Anomalies (ISSVA), 2018, GSD can be found in the section on simple vascular malformations IIa—Lymphatic malformations among other diseases (for example, generalized lymphatic anomaly (GLA), and kaposiform lymphangiomatosis, etc.), which helps in the determination of the differential diagnosis [4]. Most lymphatic abnormalities appear to be caused by somatic *PIK3CA* mutations, except in patients with kaposiform hemangioendothelioma, with or without Kasabach-Merritt phenomenon, GLA, and GSD [5].

GSD is a multifactorial disease caused by genetic predisposing factors or mosaicism for a somatic mutation [6]. Among the genes involved in various stages of pathogenesis are: *PTEN* (phosphatase and homologous tensin), *TNFRSF11A* (TNF receptor superfamily member 11a), and *TREM2* (triggering receptor expressed on myeloid cells 2) [7].

Symptoms depend on the location of the osteolytic process and present as: localized pain (the most common symptom), swelling, weakness and functional impairment of affected limbs, respiratory distress and failure, neurological defects, deformity, paralysis, and death [6].

Our aim is to hallmark the particularities of a case of GSD with multiple involvement (axial and appendicular skeleton, and spleen nodules), diagnostic difficulties, and a fatal outcome.

## 2. Case Report

An 18-year-old Caucasian female, with no family history of bone disease, was admitted to the infectious diseases unit with weakness, myalgia, significant bone pain, and febrile syndrome. A lumbar puncture excluded meningeal syndrome. After five days, an incomplete motor deficit was installed in the upper limbs and the patient was transferred to the neurosurgery service. Subsequently, the patient had multiple hospital admissions in different medical departments (oncology, neurosurgery, and the neuromotor recovery unit) and was evaluated by a multidisciplinary team in order to establish a definite diagnosis, neuromotor recovery, and neurosurgical treatments required due to complications over time.

The chest X-ray requested by the neurosurgery department showed osteolytic lesions in C6-C7, T1, and the posterior arch of the first left rib and left clavicle. Computed tomography (CT) and cervical-dorsal magnetic resonance imaging (MRI) examination revealed C6-C7-T1-T2 osteolytic lesions and C7 pathological fracture (Figure 1A).

The tumor markers carbohydrate antigen (CA) 15-3, CA 19-9, CA 125, carcinoembryonic antigen (CEA), and monoclonal antibody (CAL) had values within the normal range. Normal values of serological markers of autoimmunity, creatine kinase-MB (CK-MB), rheumatoid factor, antithyroglobulin antibody, and circulating immune complexes (CIC), excluded autoimmunity. The laboratory tests showed no increase in inflammation markers at onset, but subsequently a nonspecific inflammatory syndrome was present with a high erythrocyte sedimentation rate (ESR) of 80 mm/hour, and C-reactive protein (CRP), and fibrinogen at levels of 61.8 mg/L and 527 mg/dL, respectively.

The patient had normal fasting bone turnover markers (bone-specific alkaline phosphatase (BSAP), alkaline phosphatase, and osteocalcin) with only the beta isomer of C-terminal telopeptide of type 1 collagen (β-CTX) slightly increased (0.68 ng/mL, normal range ≤ 0.573 ng/mL). Bone mineral metabolism markers, ionized calcium, magnesium, phosphorus, and parathyroid hormone (PTH), were normal except for 25-hydroxy vitamin D, which was low (15.4 ng/mL). The results of the main laboratory tests are summarized in Table 1.

A bone marrow biopsy showed osteolysis, replacement of bone tissue with connective tissue, and chronic non-specific inflammation (Figure 2).

The diagnosis of Gorham-Stout disease was established based on the clinical and imaging elements, as well as the exclusion of pathologies such as thyroid or parathyroid disease, lymphoma, neoplasia, and autoimmune disorders.

The evolution was unfavorable and rapidly progressive with a severe motor deficit in the lower limbs and cervicothoracic pain. The CT scan and the MRI examination showed the mechanical deterioration of the previous fixation system, cervical spine dislocation with maximum angulation angle at the C6-C7 level, and spinal stenosis at the C6-C7 level (Figure 1B–D and Figure 3).

The main complications included rapid and severe neurological deficits, chylothorax and mechanical damage to the anterior fixation system, and, over time, damage of the osteosynthesis material.

The surgeries, performed in three steps, are summarized in the Table 2.

Pleural effusion appeared in the first months and gradually increased in quantity. A puncture was performed and the results of biochemical and cytological tests showed the presence of an exudate with a predominance of lymphocytes: albumin 2.66 g/dL, glucose 103 mg/dL, lactate dehydrogenase (LDH) 178 U/L, protein 4.67 g/dL, sediment, and cell block with moderate cellularity consisting of small lymphocytes.

The osteolysis was not limited to the bones affected initially. Pelvic X-ray (anteroposterior view) revealed new osteolytic areas in the lumbar vertebrae, pelvic bones, and the bilateral femur (Figure 4A). An abdominal CT showed a normal sized spleen, but with multiple splenic nodules (Figure 4B).

The neurological deficit was progressive to spastic paraplegia, an inability to walk and stand. The patient received recovery therapy during hospitalizations in the neuromotor recovery clinic and pharmacological treatment, but with modest results (Table 3).

Unfortunately, death occurred due to neurological complications in association with the presence of chylothorax that generated an impairment of the respiratory function.

This study was approved by the Ethics Committee of Clinical Rehabilitation Hospital Iasi (approval no. 24247/4.11.2020) and was conducted in accordance with the Declaration of Helsinki. Written informed consent was obtained from the patient at the second hospital admission.

## 3. Discussion

GSD is a rare disease and has no gender or race predilection [8]. Rare diseases affect less than 5 individuals per 10,000 people [9].

There are no specific tests or biomarkers for the diagnosis of GSD. The positive diagnosis imposes the exclusion of conditions such as thyroid or parathyroid disorders, cancer, lymphoma, autoimmune disorders, and others syndromes with massive osteolysis. Numerous hospital admissions in different departments like neurosurgery, neuromotor recovery unit, and oncology, were necessary in order to complete the diagnosis. Different tests were performed and a multidisciplinary approach was applied. The diagnosis was established seven months after the onset.

Heffez established a set of diagnostic criteria for GSD: (1) biopsy showing angiomatous tissue or fibrous connective tissue; (2) absence of cellular atypia; (3) minimal or no osteoblastic response and absence of dystrophic calcifications; (4) evidence of progressive local bone resorption; (5) the lesion is not ulcerative and does not provoke cortical expansion; (6) absence of visceral involvement; (7) osteolytic radio-graphic pattern; and (8) negative hereditary, metabolic, neoplastic, immunological, and infectious etiology [7,10].

In our case, the first two criteria were met because the bone marrow biopsy showed osteolysis, replacement of bone tissue with connective tissue, and no sign of cellular atypia. The patient showed minimal osteoblastic response because she had normal fasting bone turnover markers with only β-CTX slightly increased. Normal bone mineral metabolism markers and the absence of dystrophic calcifications are the evidence for the third criterion. Local bone resorption was progressive. The onset of lytic lesions were described in the left clavicle and, in just a few months, the imaging examinations showed almost complete osteolysis of the left clavicle with an absence of the acromial end. The cortex of the bone was affected by destruction and resorption, without cortical expansion, which is consistent with criterion 5. There was no visceral damage, except the splenic nodules seen on the abdominal CT scan. The association between characteristic progressive bone lesions and splenic and soft tissue lesions is pathognomonic for GSD [11].

Four stages were described in terms of radiological appearance. In the first stage there were intramedullary and subcortical radiolucent foci, and in the second stage they converged and extended to the periphery of the bone. In the third stage cortical erosion and invasion of adjacent soft tissues appeared. In the last stage, the bone fragment involved disappeared [12,13]. MRI elements in adults with GSD can be hypointensity and hyperintensity in T1 and T2 weighted images [14]. In children under 10 years of age, T1 hyperintensity is due to subcutaneous fat [14,15]. In our patient’s case, within a few months of the onset, there were radiological lesions in all stages: stages I-III (spine), stages I-II (ribs, pelvic bones, and femur), and stage IV (clavicle). The negative family history and the normal results of the laboratory tests excluded other pathologies (thyroid or parathyroid disease, lymphoma, neoplasia, and autoimmune disorders), thus fulfilling the last criterion proposed by Heffez.

### 3.1. Clinical Features

The clinical picture is heterogeneous in GSD and depends on the involved structures and the progression of osteolysis [16]. The average age of onset is 25 years, but age extremes can be reached (from 1 month to 75 years of age) [6,17]. In some cases, a previous trauma has been described [18]. In our case, the onset of disease was before the age of 20, and there was no history of trauma.

Symptoms presented in GSD include localized pain, swelling, weakness and functional impairment of affected limbs, respiratory distress and failure, neurological defects, deformity, and paralysis [6]. The clinical features of osteolysis are non-specific, and often GSD patients with vertebral involvement enter medical service when a pathological vertebral fracture or a symptomatic vertebral deformity occurs [19,20]. At the time of onset, our patient had a pathological fracture of the C7 vertebra and the manifestations were significant: bone pain, weakness, myalgia, febrile syndrome, and an incomplete motor deficit in the upper limbs.

Osteolysis can affect both the axial skeleton (cranium, facial bones, rib cage, sternum, and the vertebral column) and the appendicular skeleton (shoulder girdle, pelvic girdle, upper and lower extremities). Theoretically any bone can be affected, but it seems that the most commonly involved are: maxillofacial region in 30% of cases (especially the mandible), clavicle, ribs, cervical vertebrae, pelvis and femur [6,16].

A comparative study between GSD and GLA (two entities which are similar and, thus, can make differential diagnosis a challenge) showed that the appendicular skeleton is involved in only 26.3% of patients with GSD compared to patients with GLA (87.5%). The ribs were the bones most frequently involved in both situations, followed by the skull, clavicle, and cervical spine in GSD, and, in GLA, the thoracic spine, humerus, and femur [21,22]. In GSD pathological fractures are more frequent than in GLA [23]. In GSD one or more bones are affected with progressive cortical osteolysis, visceral thoracic, or abdominal damage, and pleural effusions [24]. In a literature review conducted by Fares et al., only 37 cases of GSD with shoulder girdle damage are mentioned, with the following location correlated with the girdle elements: humerus (54%), scapula (35%), and clavicle (30%) [25]. In our patient’s case, osteolysis foci were multiple, both in the axial (cervical, thoracic, and lumbar spine, and ribs) and appendicular (clavicle, pelvic bones, and femur) skeleton.

### 3.2. Pathogenesis

GSD is characterized by massive progressive osteolysis and vascular proliferation [2]. Several main hypotheses were proposed in the etiopathogenesis of GSD: local hypoxia/acidosis, endothelial dysplasia, increased osteoclastic activity or sensitivity of osteoclast precursors to humoral factors, lack of thyroid C cells and calcitonin, and lymphatic vessel proliferation [7].

In very recent original research, Rossi et al. (2020) observed that 75% of osteoclasts of patients with GSD presented suggestive elements for a more motile phenotype (lamellipodia, stress fibers, and membrane ruffling) and an increased activity of more than 4.5 times [26]. Transcriptomic analysis by the authors showed enrichment of some pathways involved in osteoclasts differentiation and function (angiotensin II-stimulated signaling through G proteins and beta-arrestin, the PI3 kinase pathway, and the EGF receptor signaling pathway) [26].

At the level of osteoblasts in patients with GSD, it is an increase of gene expression for *MMP13* (matrix metallopeptidase 13; the gene product is involved in capillary formation and osteocytic perilacunar remodeling) and *RXFP1* (relaxin family peptide receptor 1; its overexpression in fibrocartilaginous cells causes increased matrix metalloproteinase-13 and 9) [26].

There are no biomarkers available for GSD, but in some studies miRNAs and serum biomarkers were reported as possible biomarkers. There are studies that show changes in miRNAs in GSD, suggesting that these molecules could be used as potential biomarkers in GSD [27]. Some of these molecules of miRNAs (miR-1246, miR-137, and miR-1) are correlated with angiomatous proliferation and regulation of osteoclastogenesis and altered osteoclast morphology resulting in a more motile phenotype [27]. Other miRNA molecules (miR-204-5p, miR-378a-3p, miR-615-3p, and miR-204-5p) play a role in inhibiting osteoblast differentiation and are elevated in GSD [27]. Other potentially relevant and high serum biomarkers in GSD are pyridinoline cross-linked carboxy-terminal telopeptide of type I collagen (ICTP; reflects increased bone resorption activity), sclerostin (associated with defective bone regeneration), VEGF-A (sign of angiomatous proliferation) and IL-6 (stimulates osteoclastogenesis and angiogenesis) [26,28].

The characteristics of osteolysis are spontaneous, idiopathic, and progressive without new bone production [29]. Histopathological lesions are represented by massive loss of the bone matrix, and proliferation of thin-walled capillary-sized vascular channels or fibrous connective tissue [7,17,30,31]. It was observed that progressive bone resorption (partial or complete) has not always been followed by significant vascular proliferation [30,32]. The lesions are progressive and can spread to nearby bones or adjacent structures [30].

The natural evolution of bone lesions is difficult to predict. Bone resorption can stop spontaneously and this it is the most common evolution. There are some cases where lesions have remained stable for decades without any reossification [17,33].

### 3.3. Treatment

There are no standardized treatment protocols, given the rarity of the disease, but there were trials of pharmacological therapy, surgery, radiotherapy or combinations therein [34]. Pharmacological treatments used included: interferon, bisphosphonates, calcium salts and vitamin D; interferon and zoledronic acid; cyclophosphamide and fluorouracil; and salmon calcitonin, alendronate sodium, and sirolimus [26,34]. Sirolimus is a mammalian target of rapamycin (mTOR) inhibitor, which has antiangiogenic activity and the effect of improving clinical symptoms and quality of life in GSD patients [35]. Rossi et al. suggested that pharmacological treatments should include an anti-osteoclast drug, anabolic drug, and angiogenesis inhibitors [26]. The surgical techniques (bone grafting reconstruction, amputation, and massive prosthetic reconstruction) are usually associated with complications such as infections, nerve and vascular damage, recurrence of osteolysis, and graft resorption [36]. For chylothorax, surgical treatment includes thoracic duct ligation, pleurodesis, and chest drainage [37]. Radiation therapy may be effective in preventing disease progression [38]. Surgical treatment and radiation therapy are preferred in cases with extensive lesions and functional instability [2].

In our case, the treatment was a combination of surgery, pharmacological, and recovery therapy. Initially, a corpectomy and reconstruction was performed with autologous graft and fixation with a cervical-thoracic hybrid system, but in time mechanical damage appeared to the anterior fixation system and, later, damage of the osteosynthesis material. It involved surgical reintervention with reconstruction and fixation with a hybrid system. Recovery therapy (physiotherapy, massage, and kinetotherapy) and pharmacological treatment brought only mediocre short-term improvements. Chest drainage was performed for chylothorax.

### 3.4. Life Expectancy, Complications, Mortality

In most cases the progression of the disease is slow, but the prognosis remains unpredictable [29]. Involvement of only the limbs or pelvis does not influence life expectancy [33,39]. The involvement of certain bones such as ribs, scapula, and thoracic vertebrae can cause chylothorax and nerve root compression with paraplegia, and are considered negative prognostic factors because life-threatening complications can occur [16,17,40].

Chylothorax has been identified in 17% of cases with thoracic involvement [17]. The mechanisms for pleural effusion include the direct extension of the disease into the pleural cavity or thoracic duct involvement [11].

Overall mortality was estimated at 13% [29]. Mortality increased to 30% when osteolysis involved the spine, with possible risk factors in this case being spinal cord compression or transection of the spine caused by vertebral instability secondary to osteolysis [17,41]. Mortality increased to 43.6–50% in cases of chylothorax [15,41].

The unfavorable prognostic factors in our case were the involvement of cervical and thoracic vertebrae, ribs that produced chylothorax, nerve root compression with paraplegia, and, eventually, respiratory failure and death.

## 4. Conclusions

Gorham-Stout disease is a multifactorial one, which may have a poor prognosis and should be considered after the exclusion of other pathologies with massive osteolysis. Osteolytic lesions of the spine and pleura are poor prognostic factors because of the compression of spinal nerves and the presence of chylothorax.

## Figures and Tables

**Figure 1 medicina-57-00681-f001:**
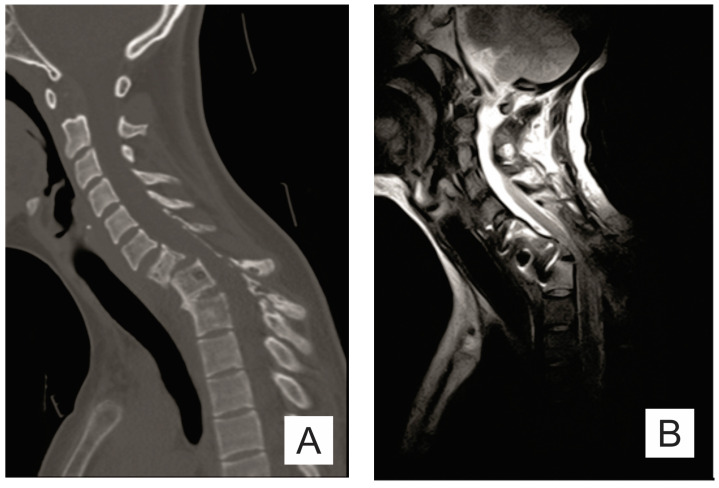
(**A**) Sagittal CT scan: osteolytic lesions (C6–C7, T1–T2), pathological fracture of the C7 vertebra; (**B**) sagittal T2 MRI; (**C**) sagittal STIR MRI; and (**D**) sagittal T1 MRI: important spinal canal stenosis at C6–C7 level up to an anteroposterior diameter of 5 mm, the medullary cord presents in this area discreetly T2-STIR hypersignal.

**Figure 2 medicina-57-00681-f002:**
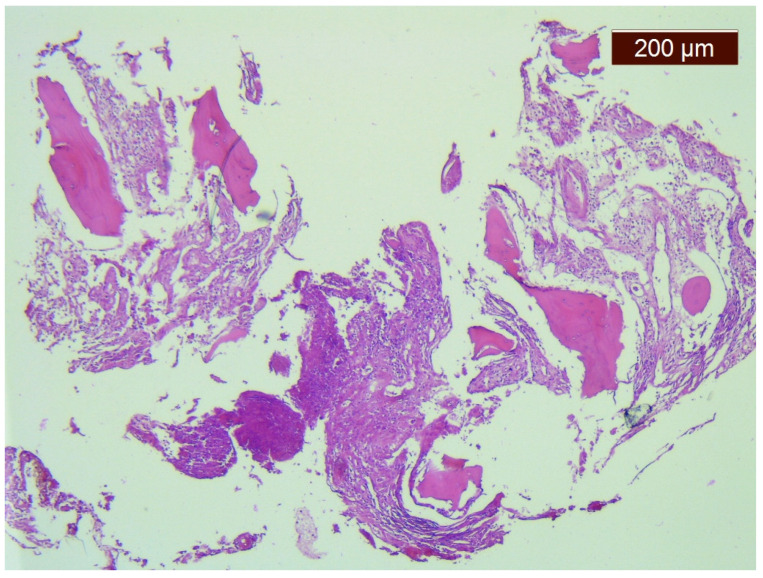
Microscopical aspect of bone marrow biopsy showing fibrosis, foci of osteolysis, and moderate inflammatory infiltrate with lymphocytes and plasma cells (HE, ×50).

**Figure 3 medicina-57-00681-f003:**
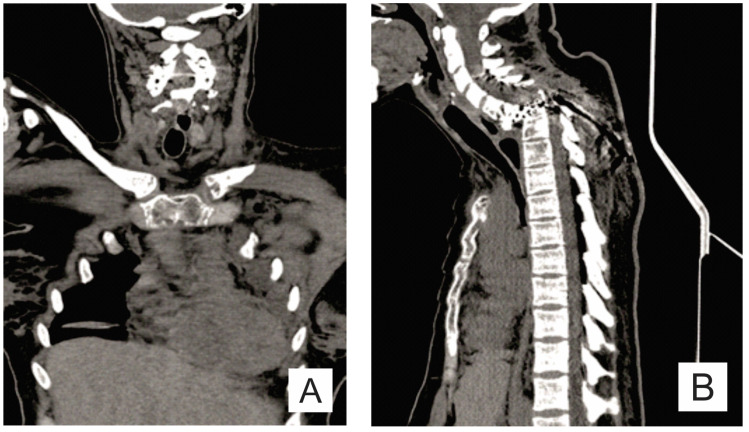
(**A**) Coronal CT scan: almost complete osteolysis of the left clavicle with absence of acromial end; and (**B**) sagittal CT scan: cervical spine dislocation with maximum angulation angle at the C6–C7 level, spinal cord compression at this level, and bone lysis in the C6 and C7 bodies.

**Figure 4 medicina-57-00681-f004:**
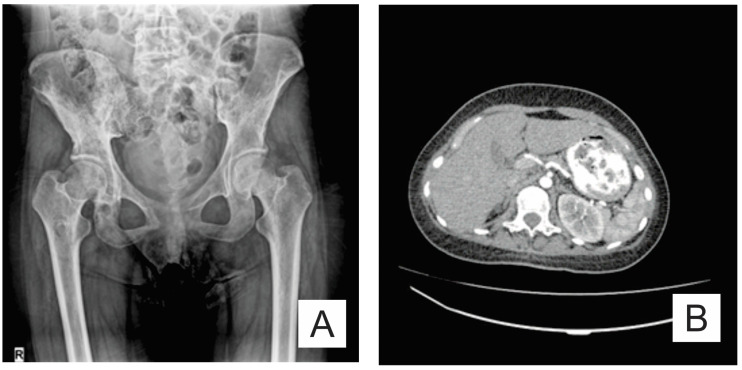
(**A**) Pelvis X-ray (AP view): osteolysis areas in the lumbar vertebrae, pelvic bones and the bilateral proximal femur; and (**B**) axial CT scan: multiple splenic nodules.

**Table 1 medicina-57-00681-t001:** Laboratory tests.

Laboratory Test	Reference Value	Patient Data	Laboratory Test	Reference Value	Patient Data
Calcium	4.61–5.17 mg/dL	4.98	CA 15-3	0–32.4 U/mL	12.4
Magnesium	1.6–2.5 mg/dL	1.94	CA 19-9	0–30.9 IU/mL	7.29
PTH	15–65 pg/mL	18.22	CEA	0–2.5 ng/mL	<0.01
Procalcitonin	0.05–0.5 ng/mL	<0.12	CA 125	1.9–16.3 U/mL	11.3
25-hydroxy vitamin D	30–100 ng/mL	15.4	CK-MB	0–24 U/L	8.92
β−CTX	≤0.573 ng/mL	0.68	Rheumatoid factor	<14 IU/mL	6
BSAP	3–19 μg/L	11.6	Antithyroglobulin antibody	≤115 IU/mL	13.58
Osteocalcin	11–43 ng/mL	13	Anti-dsDNA antibody	<100 IU/mL	<10
Alkaline phosphatase	204.59 U/L	0–800 U/L	Antinuclear antibody	<1/80	<1/80
Fibrinogen	200–400 mg/dL	527 mg/dl	CIC	<20 RU/mL	5.404
ESR	2–20 mm/hour	80	ACE	12–68 U/L	45
CRP	0.1–4.1 mg/L	61.8	CAL	<9.82 pg/mL	0.528

PTH: parathyroid hormone; β-CTX: beta isomer of C-terminal telopeptide of type 1 collagen; BSAP: bone-specific alkaline phosphatase; ESR: erythrocyte sedimentation rate; CRP: C-reactive protein; CA: carbohydrate antigen; CEA: carcinoembryonic antigen; CK-MB: creatine kinase-MB; CIC: circulating immune complexes; ACE: angiotensin-converting enzyme; and CAL: monoclonal antibody.

**Table 2 medicina-57-00681-t002:** Surgical treatment.

Time	Surgical Treatment
Month I	C7 corpectomy and reconstruction with autologous graft from the iliac crest, macroscopic resection of the posterior arches C7-T1 and fixation with C4-C5-T3-T4 cervico-thoracic hybrid system
Month III	Ablation of osteosynthesis material and bone graft and T1 corpectomy, reconstruction with C6-T1 mesh fixed with proximal and distal screw
Month IX	Ablation of damaged osteosynthesis material mesh C6-T1, ablation of thoracic screws, and introduction of bilateral T3, T4, and T5 screws

**Table 3 medicina-57-00681-t003:** Neurological deficits and recovery treatment over time.

Time	Neurological Deficits	Neuromotor Recovery Treatment
Month IV	Lower limb motor deficit (MRC 1/5 proximal and 0/5 distal), bilateral pyramidal syndrome (elastic hypertonia Ashworth 3, brisk DTR, absent SAR, positive Babinski sign bilateral, left Achilles tendon clonus), and inability to walk, but possible orthostatism for a short time with assisted support	physiotherapy, massage, kinetotherapy, pentoxifylline, gabapentin, baclofen, alfacalcidol, vitamin D and calcium supplements
Month VII		
Month XII	Lower limb motor deficit (MRC 1/5 proximal and 0/5 distal), bilateral pyramidal syndrome (elastic hypertonia Ashworth 3, brisk DTR, absent SAR, positive Babinski sign bilateral, left Achilles tendon clonus), and inability to stand and walk	

MRC: Medical Research Council scale; DTR: deep tendon reflexes; and SAR: superficial abdominal reflex.

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
