# Peer review of "Gorham-Stout Disease with Multiple Bone Involvement—Challenging Diagnosis of a Rare Disease and Literature Review"

_medicina, 2021, doi:10.3390/medicina57070681_

Round 1

Reviewer 1 Report

The paper is interesting and well written, but same papers should be cited and discussed:

FASEB J. 2021 Mar;35(3):e21424. doi: 10.1096/fj.202001904RR.

Bone. 2020 Jan;130:115068. doi: 10.1016/j.bone.2019.115068. 

J Bone Miner Res. 2006 Feb;21(2):207-18. doi: 10.1359/JBMR.051019.

Author Response

1. The paper is interesting and well written, but same papers should be cited and discussed:

FASEB J. 2021 Mar;35(3):e21424. doi: 10.1096/fj.202001904RR.

Bone. 2020 Jan;130:115068. doi: 10.1016/j.bone.2019.115068. 

J Bone Miner Res. 2006 Feb;21(2):207-18. doi: 10.1359/JBMR.051019.

Response 1:

Thank you very much for these comments on the paper.

We have entered comments on the articles mentioned in the pathogenesis section.

Reviewer 2 Report

Momanu et al. described a patient with Gorham-Stout disease and performed review of literature. Since GSD is a very rare disease, the increase of cases description is important to better understand the etiology of the disease.

The paper is insteresting. There are issues that should be modified/inserted:

  1. The Figure 1 can be removed since it does not add further information
  2. The histological pictures of bone biopsy must be inserted; ematoxylin/eosin and/or TRAcP stainings to identify osteolytic areas and osteoclasts should be inserted. Moreover podoplanin staining to evaluate lymphatic vessel should be added.
  3. Did the authors observe bone marrow fibrosis in bone biopsy of patient?
  4. To complete the literature review, the authors should insert two papers recently published on GSD, Rossi M et al., 2020 and Rossi et al. Faseb 2021. These two papers can be discussed in the pathogenesis section.
  5. In the treatment, they should cite the off-label use of sirolimus as shown by some published papers.

Author Response

Thank you very much for these comments on the paper.

  1. The Figure 1 can be removed since it does not add further information

Response 1: Figure 1 was replaced with histological picture of bone biopsy.

  1. The histological pictures of bone biopsy must be inserted; ematoxylin/eosin and/or TRAcP stainings to identify osteolytic areas and osteoclasts should be inserted. Moreover podoplanin staining to evaluate lymphatic vessel should be added.

Response 2: Histological picture of bone biopsy with Hematoxylin and eosin staining was added.

The diagnosis of GSD was made at the seventh month after the onset after the exclusion of neoplastic processes, infections, endocrine and metabolic diseases following multiple hospitalizations in various medical services in different hospitals. The biopsy was performed in the second month after onset. Hematoxylin and eosin staining was performed at that time, but not podoplanin staining (which is a not usually staining) because no GSD hypothesis was at that time.

  1. Did the authors observe bone marrow fibrosis in bone biopsy of patient?

Response 3: The pathological result describes the fact that there is fibrosis (added in figure 1 description).

  1. To complete the literature review, the authors should insert two papers recently published on GSD, Rossi M et al., 2020 and Rossi et al. Faseb 2021. These two papers can be discussed in the pathogenesis section.

Response 4: We completed the literature review with the two suggested papers. The information was introduced in the pathogenesis section.

  1. In the treatment, they should cite the off-label use of sirolimus as shown by some published papers.

Response 5: In the treatment section we have added references to the benefits of treatment with Sirolimus.